# Predictors of Low Back Pain Risk among Rubber Harvesters

**DOI:** 10.3390/ijerph191710492

**Published:** 2022-08-23

**Authors:** Parnchon Chokprasit, Supabhorn Yimthiang, Siriluk Veerasakul

**Affiliations:** 1Department of Environment Technology Safety and Health, School of Public Health, Walailak University, Nakhon Si Thammarat 80161, Thailand; 2Research Center of Workers Health, Walailak University, Nakhon Si Thammarat 80161, Thailand; 3Center of Excellence in Data Science for Health Study, Walailak University, Nakhon Si Thammarat 80161, Thailand

**Keywords:** risk factor, low back pain, rubber farmer, harvesting, Thailand

## Abstract

Low back pain (LBP) is a significant work-related musculoskeletal disorder among rubber farmers. This major occupational health problem was highly reported in the agricultural sector. While rubber farming is a profession with high risk of LBP, predictors for LBP remain unclear. This study was designed to investigate the risk predictors of LBP among rubber farmers during the harvesting process. A cross-sectional study was conducted between January and March 2021, in which an interviewer administered a pretested structured questionnaire. Bivariate and multivariate binary logistic regression analyses were performed. A total of 317 rubber farmers were recruited with a 100% response rate. The prevalence of LBP was 71.2% with 95% confidence interval (CI) of (0.716–1.900). Significant risk predictors were working experience (adjusted odds ratio (AOR): 1.743, 95% CI (1.034–2.937)), agricultural registration (AOR: 2.022, 95% CI (1.078–3.792)), work without training (AOR: 2.037, 95% CI (1.083–3.832)), heavy workload (AOR: 2.120, 95% CI (1.242–3.621)), and prolonged standing (AOR: 2.944, 95% CI (1.586–5.465)). Intriguingly, those with sufficient income had a reduced risk of LBP than those with insufficient income. This study confirmed that LBP is a major work-related musculoskeletal disorder among rubber farmers. The result here suggests that the five predictors reported above should be prioritized for further disease prevention.

## 1. Introduction

Low back pain (LBP) is a chronic illness with a high rate of occurrence and a high degree of severity [1]. LBP among farmers is one of the top three high-risk occupational diseases that greatly affects the workforce [2]. LBP is one of the important public health problems that can cause work absenteeism and increase occupational disability costs. In the agricultural sector, LBP disrupts normal work in 27.90% of farmers [3]. LBP is a major occupational health problem worldwide. LBP prevalence is approximately 50% or higher among dairy farmers in Sweden [4], rice farmers in India [5], and oil palm harvesters in Malaysia [6]. In Sri Lanka, 94.4% of rubber farmers exhibited work-related LBP [7]. Compared to individuals who have worked for less than 2 years, employees who have worked for 2 years or more are more likely to develop LBP [8]. In Thailand, LBP accounts for 31% of all occupational diseases. LBP results in pain, swelling, fatigue, stiffness, inflammation, numbness, and limited movement. The tendency to develop such pain depends on the nature of the farmer’s job [9,10]. LBP is defined as pain from work, which is influenced by muscle tension, or stiffness of the area between the inferior margin of the 12th rib and inferior gluteal folds with or without sciatica [11]. Extreme physical activities direct all the stress to the spinal nerves inside the vertebrae, resulting in inflammation of the bones and muscles [12]. Farmers plant and harvest their products for 10 to 12 h daily, which includes time outside the normal working hours to earn extra income [13,14]. Long labor hours with twists and awkward postures were the main cause of LBP in Swedish farmers [15]. Heavy workload has resulted in disorders of tendons, ligaments, joints, nerves, blood vessels, bone, and muscle, all of which are directly caused by working posture and repetitive movement. It is the cause of back pain lasting at least 6 months, increasing the direct stress on the body [8,16,17]. LBP in farmers may be influenced by a variety of individual and ergonomic factors [18]. The incidence of work-related musculoskeletal disorders, including LBP, is reportedly influenced by personal characteristics such as age, sex, marital status, education, job experience, body mass index (BMI), smoking, income, expenditures, and non-work-related exercise [18,19]. Psychosocial factors, such as anxiety and stress [13], as well as break time during the workday [20] also play a role. Due to the demand from the market, farmers have a heavier workload during the harvest. Therefore, they were more likely to be exposed to ergonomic problems such as kneeling, squatting, heavy lifting, holding the same posture for longer than 8 h, prolonged walking and standing, expending more effort than usual, heavier workload, and the stress of movement directly on the body [16,17]. As previously noted, farmers are likely to be at significant risk of developing LBP and other work-related musculoskeletal diseases during harvest. However, the precise risk factors associated with LBP are unclear and vary depending on occupational and other environmental factors.

Para rubber is an important economic crop of Thailand that produces its top global export product [21]. While the number of rubber farmers is growing, this profession is susceptible to LBP. Epidemiological studies of LBP prevalence among these workers are limited. Most of the studies used musculoskeletal disorder as an outcome variable. Furthermore, little is known regarding the job profile, ergonomics, and psychological risk [22] factors for LBP among rubber farmers. Rubber tapping is the process by which natural rubber is collected. When the circumference of the tree trunk reaches 50 cm, tapping can occur. The tapping level usually occurs at a height of 150 cm above the ground and then moves down nearly to ground level. After that, the next part of the tree trunk is tapped at a level of 150 cm. A tapping knife is used to cut the tree bark downward at a 30-degree angle along a left-to-right oblique curve that cuts through the latex vessels. Most of the rubber farmers are also exposed to ergonomic hazards since their job entails repetitive tapping movements, which is often repeated hundreds of times per day in awkward postures of the upper limbs, shoulder, neck, trunk, knee, and legs [23,24]. In the process of harvesting the crops, rubber farmers need to be in other awkward postures such as bending down, twisting, turning the body to lean 15–30 degrees along the rubber tree, squatting in the low front tire tapping, standing, and walking for a long time according to the number of rubber trees that are about 700 per 8000–16,000 m^2^, lifting and moving heavy products (Figure 1). These activities lead to both acute and chronic bodily injuries [25].

The observations above indicate that rubber farmers are a key workforce in Thailand, who exhibit occupational health risks, particularly during the harvesting period. These workers constantly face LBP risks and continue to report the occurrence of the disease. LBP development varies according to occupation and environmental factors. Predicting the risk factors of disease progression is likely to be beneficial for further disease prevention. Therefore, this study aimed to investigate the prevalence of LBP and its risk factors among rubber farmers during the harvesting process. We hypothesized that the subject characteristics, working profile, and ergonomic risk variables could predict the trends of symptom severity on LBP occurrence among rubber farmers.

## 2. Materials and Methods

### 2.1. Study Design and Setting

A cross-sectional study was conducted among rubber farmers, aged 18–60 years, in Tha Khun Sub-district, Taling Chan Sub-district, Moklan Sub-district, Tha Sala District, Nakhon Si Thammarat Province. The study recruited the largest number of registered rubber farmers in these areas. In this study, the risk factors for LBP were investigated, including sociodemographic/individual characteristics including those of sex, age, education, marital status, BMI, smoking, working experience (years), physical activity, income based on household consumption expenditures of southern Thai people, and stress/anxiety. Working profiles include working days off per week, working hours per day, break time in a day, inherited careers, overtime, agricultural registration with the Ministry of Agriculture and Cooperatives, work without training, and ergonomic factors such as repeated movement, squatting, kneeling, heavy workload (continuous harvesting for over 7 h), getting muscle strain from the body directly, heavy lifting (lifting and carrying latex buckets weighing approximately 10 and 60 kg continuously) were also investigated.

### 2.2. Sample Size Calculation and Sampling Technique

The sample of 317 rubber farmers was calculated using the formula of WG Cochran (1953), in which 0.0519 percentage of error was conducted [26]. Subjects were selected using multi-stage random sampling. Using purposive random sampling, the three sub-districts with the highest number of rubber farmers were selected, then proportional sampling was based on the number of rubber farmers. A sample using probability theory was used to obtain a sample for each district. Rubber farmers aged 18–60 years, who have worked in rubber tapping for 2 years or more, were included [8]. To obtain the most accurate information and minimize recall bias, these subjects carried on with their regular work activities for the 28 days prior to data collection. The subjects had never been diagnosed with work-related musculoskeletal disorders (WMSDs). Those with current musculoskeletal trauma were excluded from the study.

### 2.3. Data Collection Tools and Procedures

Face to face interviews, as well as observational analysis of various tasks, were employed. The questionnaire consists of sociodemographic/individual factors, working profile and, ergonomics risk factors. The Nordic Musculoskeletal Questionnaire (NMQ) was adopted to investigate low back pain [27,28]. Then, the pain score at greater than or equal to 3 out of 10 on the (Numeric rating scale: NRS) was categorized as having LBP. Different combinations of validated and standardized questionnaires were used for collecting different types of data. The questionnaires were proved valid by 3 experts with an inspection content validity of 0.95 and were tested for reliability with Cronbach’s alpha was 0.89.

### 2.4. Data Analysis

The Chi-squared test was used to analyze the influence of sociodemographic/individual characteristics, working profile, and ergonomic factors among the different subgroups with and without LBP. The logistic regression model was used to identify the associated factors of LBP. Independent variables with a *p*-value of less than 0.05 in the bivariate logistic regression were taken into the multivariate logistic regression to control the possible main effect of confounders [29]. The data were reported using an adjusted odds ratio (AOR). The 3-marker (sociodemographic/individual predictors, working profile predictors) and 4-marker models (ergonomics risk factor) were conducted to examine LBP risk factors in rubber farmers. The Hosmer-Lemeshow test was used to confirm that the models were accurate. Finally, variables with *p* < 0.05 in the multivariate analysis were considered statically significant and presented using AOR and 95% confidence interval (CI). The research framework depicting the variables and the potential associations are presented. OR > 1 was considered a risk factor toward LBP [30]. Statistical analysis was performed using IBM SPSS Statistics28.0 (IBM Site Number: 4225416, IBM Corp., Armonk, NY, USA).

### 2.5. Ethics Consideration

The experimental study was approved by the ethical review committee of Walailak University (WUEC-20-366-01). All of the subjects participated voluntarily and were asked to sign informed consent after receiving the details of the study. The authors have no conflict of interest related to the study.

## 3. Results

### 3.1. Subject Characteristics

A total of 317 rubber farmers participated in the study (100% response rate). The farmers were divided into two groups: those with and without LBP. At least one symptom of LBP was present in 71.29% of the subjects. Working for over 20 years, inadequate income, stress, and anxiety were significantly associated with LBP (Table 1). Working profile, including heirloom agricultural land, registered farmers, and habitual work, were significantly associated with LBP (Table 2). Ergonomic risk factors were also linked to LBP (Table 3). The working postures and the duration of work influenced the symptoms of LBP. In this regard, rubber farmers who worked with prolonged squatting, kneeling, heavy workload, direct muscle tension, and prolonged standing showed LBP symptom development.

### 3.2. Predictors of Low Back Pain (LBP) among Rubber Harvesting

#### 3.2.1. Multivariate Analysis

Factors associated with LBP were analyzed using bivariate logistic regression. Eleven univariately associated factors with a *p*-value of less than 0.01 and 0.05 of LBP were utilized in the multivariate logistic regression analysis to develop a risk model. The sociodemographic/individual characteristics, working profile, and ergonomic risk factors in the bivariate were taken into the multivariate analysis. The multivariate binary logistic regression analysis showed a statistically significant association between risk factors and LBP. Adequate income presented as a protective factor related to LBP. Rubber farmers who have sufficient income were 56.1% less likely to develop LBP (AOR: 0.419, 95% confidence interval [CI] (0.220–0.799)). Furthermore, the factors that showed significant predictors for LBP were agricultural registration and prolonged standing. Rubber farmers who were registered (AOR: 2.218, 95% CI (1.132–4.346) were LBP 2.218-fold more likely to develop to than those without this factor. Participants who worked with prolonged standing (AOR: 2.948, 95% CI (1.546–5.618), were more likely to develop LBP than those who did not (Table 4).

#### 3.2.2. Three-Marker Model

Among sociodemographic/individual factors, working experience significantly predicted LBP risk. Participants with working experience of more than 16.76 years were 1.743-fold more likely to develop LBP (AOR: 1.743, 95% CI (1.034–2.937)). In contrast, those with sufficient income was 60.7% less likely to develop LBP (AOR: 0.393, 95% CI (0.215–0.718)). Two working profile factors, agricultural registration and working without training, also significantly predicted risk of LBP. Workers with agricultural registration was 2.022-fold more likely to develop LBP (AOR: 2.022, 95% CI (1.078–3.792)) while those with habitual work were 2.037-fold more likely to develop LBP (AOR: 2.037, 95% CI (1.083–3.832)) (Table 4).

#### 3.2.3. Four-Markers Model

Two ergonomic factors, heavy workload and prolonged standing, are predictors of LBP risk. Rubber farmers with heavy workload were 2.120-fold more likely to develop LBP (AOR: 2.120, 95% CI (1.242–3.621)). Those with prolonged standing were 2.944-fold more likely to develop LBP (AOR: 2.944, 95% CI (1.586–5.465)) (Table 4).

#### 3.2.4. Potential Value of Risk Predictors

Predictive models and their variation trends during rubber harvesting were important in LBP monitoring. Rubber farmers with the following factors during the harvesting process: agricultural registration, working without training, working experience, working with kneeling, heavy workload, directly getting muscle tension and prolonged standing, were more likely to have LBP than those without these factors. Interestingly, adequate income can be a protective factor against LBP.

The explanatory power of the models for multivariate analysis was generally high, with pseudo R^2^ [Nagelkerke] values of 0.216. The explanatory power of the model for the three-marker model of sociodemographic/individual predictors and working profile predictors and the four-markers model was low with a pseudo R^2^ [Nagelkerke] value of 0.100, 0.057, and 0.125, respectively.

Moreover, the results of Hosmer–Lemeshow test show that the *p*-values of the predicting models were 0.237 (Multivariate analysis), 0.609 (three-marker model of socio-demographic/individual predictors), 0.055 (three-marker model of working profile predictors), and 0.188 (four-marker model of ergonomics risk predictors) which indicates that the models are adequately calibrated.

## 4. Discussion

Our findings highlighted the occurrence of occupational diseases and the development of symptoms underlying ergonomic risk factors among farmers, particularly rubber farmers, who play an important role in the agricultural industry of the southern region of Thailand. This study reports that individual characteristics, working profile, and ergonomic factors are related to LBP. Our finding confirmed that 71.29% of rubber farmers who were working experience 2 years or more during the harvesting process had LBP. Agricultural registration, working without training, kneeling, heavy workload, directly getting muscle tension, and prolonged standing are risk factors while sufficient income is a protective factor of LBP. Furthermore, most of the farmers in this study were inheritance careers (74.1%), worked without training (83.6%), having more than 20 years of work experience (26.2%). These supported the notion that those who work constantly for a long time are more prone to experience greater musculoskeletal pain [31]. However, this study did not discover any relationship between LBP and individual characteristics such as sex, age, education, marital status, BMI, smoking, and physical activity. In contrast, the previous report has found the association of these factors with the occurrence of WMSD [19]. This result shows that occupational and other environmental factor were also likely contributing to the prevalence of WMSD, including LBP. The influence of individual factors on the occurrence of LBP remains to be investigated. The percentage of registered volunteers was found to be 83.3%. Agricultural registration presented as a predictive risk factor of LBP. Agricultural registrations enable health agencies to verify information about farmers’ LBP for the monitoring of healthcare services related to musculoskeletal diseases [32]. Consequently, access to government-registered agricultural health information occurs at a higher level in registered farmers than those without registration. However, the connection between farmer registration and LBP is still unclear.

In addition, our findings show the same trend as previous studies reported that 35% of rubber farmers who were working experience one year have LBP symptoms at least 24 h of pain duration [33]. However, the reported factors are different. We found that working experience, sufficient income, agricultural registration, work without training, heavy workload and prolonged standing presented as risk predictors for LBP, while they showed an active job, history of LBP, and current work position can affect LBP [33]. It is possible that our study was interested in predicting LBP incidence related to the harvesting season. Moreover, our study focused on the work profile and ergonomics factors. In our study, the rubber farmers aged 18–60 years, who have worked in rubber tapping for 2 years or more, were included. During the harvesting process, the farmers need to squat, kneel for the rubber tapping in the lower panel, and lift the shoulders and upper arms to support the latex bucket. Since the working muscles contract continuously while working for 5–8 h per day, having only 1–2 days off per week may not be sufficient for the rubber farmers to recover. Furthermore, most rubber farmers have harvesting areas of 8000–16,000 m^2^, which requires full-time strenuous work. These findings highlighted the heavy workload during the harvesting season that may lead to chronic tension in the musculoskeletal system. Prolonged muscle contractions cause nerve compression and may lead to acute or chronic injuries of the joints, tendons, and muscles [34], eventually leading to LBP (Figure 1). According to a previous report, palm oil farmers also work in awkward postures. Overloading and repetitive postures have resulted in musculoskeletal injuries, leading to LBP [35]. The Thai Burley Tobacco Farmers had a significantly higher prevalence of musculoskeletal disorders in the lower back [18]. In the United States, physical activities beyond acceptable tolerance levels causes muscle fatigue and muscle pain and is the main cause of lost working days [36]. Long heavy workload and awkward postures until exhaustion result in excessive direct body strain [37,38]. Heavy workload was also a predictive risk factor of LBP in this study. In addition to rubber tapping, farmers also had to lift and move the rubber bucket weighing 5–10 kg with their own hands. Previous studies reported that moving products weighing 10 kg or more and with 4 or more repetitions increases the risk of LBP in farmers [39,40]. Heavy workload caused a high strain and repeated stress on the muscles, increasing the risk of LBP [41]. Movement posture during harvesting, such as side diversion of the trunk and twist or bend of the lower, were associated with LBP [42]. In this study, rubber farmers exerted a lot of force in tapping and collecting latex and took a long time to cause acute or chronic LBP. These findings are consistent with the following theory in horticultural farmers which posits that short-term work with a high level of physical exertion is linked to acute LBP while working for a long period with low or moderate loads was associated with chronic LBP. New inflammation occurs every day from chronic long-term work [43]. Furthermore, heavy workload and insufficient income can contribute to stress and anxiety, which were associated with LBP. In this study, most of the subjects (42.9%) have not enough income to meet their expenditures resulting in financial concerns (55.2%). The rubber farmers (65.9%) need to increase their workload in order to have sufficient income. There was specific concerns about low-income people who want to work more to earn more, resulting in LBP symptoms [42]. From this study, it was found that most of the rubber farmers were in marital status (82.6%), with family burdens to take care, and earn money to support their family. When income is insufficient, it cuts extra rubber (28.4%) that very workload over time. Along with emotional stress, social stress, particularly concern about low income, will further increase pain as it motivates people to work harder for additional income. As a result, this type of stress results in LBP symptoms. Stress, anxiety, and strain on muscles triggered the secretion of lactic acid, causing fatigue and more severe LBP [44]. Moreover, this study found that working experience was a risk factor of LBP. Most rubber farmers with LBP have more than 20 years of work experience. Consistent with a study in Korean farmers, work experience was found to be associated with LBP [45]. Farmers who worked for more than 30 years had a lot of pain in the lower back [46]. Long-term work experience may be associated with repeated heavy workloads throughout the career, thus contributing to the development of LBP. However, the relationship between work experience and the mechanism of LBP remains unclear.

Furthermore, this finding showed that working without training and inheritance career is a risk factor of LBP. The agricultural sector has a high prevalence of LBP because farmers tend to work according to their habits regardless of the correct working posture. Most of them work continuously for many years without a retirement period which other occupations might have. There is also no starting age for farming because for most farmers, their work is a family heirloom; their income and expenditure are tied to their land [47]. Interestingly, sufficient income reduced the risk of LBP. Sufficient income may be related to an adequate quality of life and workload as this factor allows farmers to cover expenses for themselves and their families. It might be that when farmers have sufficient income, overtime work can be reduced. Previous studies have shown that farmers with low income need to work hard and overtime. Insufficient income is an important factor that causes LBP [48].

The limitation of the study includes the cross-sectional and self-reporting nature of the survey. The participants may not have recalled biased information of rubber farmers. Furthermore, participants may have underestimated their information to avoid being stereotyped by their assisting agencies, which govern the promotion and future opportunities in businesses related to rubber, an economic crop of the country. The data on pain relief medicines were also not included. Further studies of pain relief medications should be collected to analyses their relevance.

## 5. Conclusions

This study showed that LBP is a common work-related health problem among rubber farmers during harvesting. The three-marker risk model shows that working experience, agricultural registration, and working without training factors are risk predictors of LBP. The four-marker risk model showed the influence of ergonomic factors, especially heavy workload and prolonged standing on the severity of LBP. Predicting risk factors for LBP may be a process in developing strategies for the prevention and control of WMSDs in this occupation. The findings of this study should be used as information to train and educate about the appropriate work procedures for the rubber harvesting process. Furthermore, such predictive factors should be further explored for disease prevention.

## Figures and Tables

**Figure 1 ijerph-19-10492-f001:**
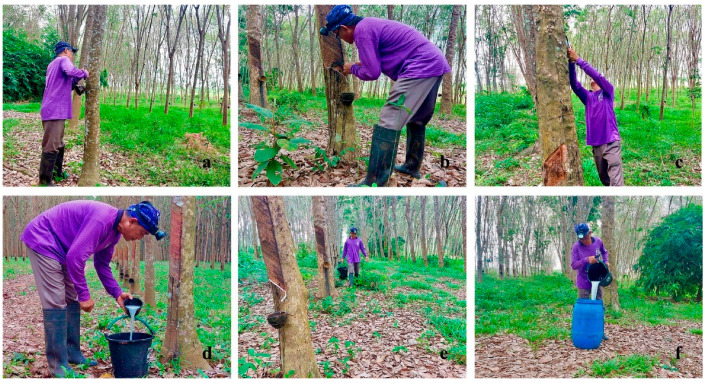
Illustrations of work postures during the harvesting process. (**a**) Body twisting along the angle of the trunk. (**b**) Bending down for rubber tapping in the low panel. (**c**) Reaching the trunk for the rubber tapping in the high panel. (**d**) Bowing down to harvest natural latex. (**e**) Lifting and moving heavy loads along the tree line. (**f**) Exerting force to support the latex bucket to pour it into a larger reservoir.

**Table 1 ijerph-19-10492-t001:** Sociodemographic/individual characteristics of LBP predictors among rubber farmers (*n* = 317).

Characteristics	Category	Total *n* = 317 (%)	Without LBP *n* = 91 (%)	With LBP *n* = 226 (%)	OR	95% Cl	*p*-Value
Sex	Male	162 (51.1)	49 (15.5)	113 (35.6)			0.535
	Female	155 (48.9)	42 (13.2)	113 (35.6)	1.167	0.716–1.900	
Age group	21–30 years	4 (1.3)	1 (0.3)	3 (0.9)			0.750
	31–40 years	45 (14.2)	12 (3.8)	33 (10.4)	0.917	0.087–9.686	
	41–50 years	130 (41.0)	34 (10.7)	96 (30.3)	0.941	0.095–9.357	
	51–60 years	138 (43.5)	44 (13.9)	94 (29.7)	0.712	0.072–7.041	
$\mathrm{Min}=28,\;\mathrm{Max}=60,\;\bar{x}$ = 48.33, S.D. = 7.201							
Educational status	Didn’t study	17 (5.4)	2 (0.6)	15 (4.7)			0.302
	Primary school	162 (51.1)	47 (14.8)	115 (36.3)	0.326	0.072–1.483	
	Secondary school	63 (19.9)	22 (6.9)	41 (12.9)	0.248	0.052–1.187	
	High school	48 (15.1)	10 (3.2)	38 (12.0)	0.507	0.099–2.590	
	Diploma	9 (2.8)	3 (0.9)	6 (1.9)	0.267	0.035–2.019	
	Bachelor’s degree	18 (5.7)	7 (2.2)	11 (3.5)	0.210	0.036–1.210	
Marital status	Single	23 (7.3)	4 (1.3)	19 (6.0)			0.406
	Married	262 (82.6)	79 (24.9)	183 (57.7)	0.488	0.161–1.480	
	Widow	19 (6.0)	6 (1.9)	13 (4.1)	0.456	0.107–1.942	
	Divorced	13 (4.1)	2 (0.6)	11 (3.5)	1.158	0.182–7.384	
Body mass index (BMI)	Underweight	207 (65.3)	53 (16.7)	154 (48.6)			0.098
	Healthy	18 (5.7)	4 (1.3)	14 (4.4)	1.205	0.380–3.820	
	Overweight	81 (25.6)	28 (8.8)	53 (16.7)	0.651	0.374–1.134	
	Obese	11 (3.5)	6 (1.9)	5 (1.6)	0.287	0.084–0.978	
$\mathrm{Min}=13.86,\;\mathrm{Max}=33.57,\;\bar{x}$ = 23.60, S.D. = 3.339							
Smoking	No	199 (62.8)	58 (18.3)	141 (44.5)			0.822
	Yes	118 (37.2)	33 (10.4)	85 (26.8)	1.060	0.639–1.756	
Working experience (years)	2–5 years	39 (12.3)	15 (4.7)	24 (7.6)			0.033 *
	6–10 years	72 (22.7)	29 (9.1)	43 (13.6)	0.927	0.417–2.060	
	11–15 years	68 (21.5)	14 (4.4)	54 (17.0)	2.411	1.007–5.770	
	15–20 years	55 (17.4)	12 (3.8)	43 (13.6)	2.240	0.903–5.556	
	>20 years	83 (26.2)	21 (6.6)	62 (19.6)	1.845	0.818–4.161	
$\mathrm{Min}=2,\;\mathrm{Max}=54,\;\bar{x}$ = 16.76, S.D. = 9.932							
Physical activity	No	161 (50.8)	54 (17.0)	107 (33.8)			0.161
	Exercise 3 days/week	81 (25.6)	17 (5.4)	64 (20.2)	1.900	1.015–3.556	
	Stretching 3 days/week	17 (5.4)	6 (1.9)	11 (3.5)	0.925	0.325–2.636	
	Household activity	58 (18.3)	14 (4.4)	44 (13.9)	1.586	0.800–3.145	
Sufficient income ^a^	Not enough(income 5000–15,000 baht)	136 (42.9)	23 (7.3)	113 (35.6)			0.001 **
	Enough but not left (income >15,001–20,000 baht)	126 (39.7)	50 (15.8)	76 (24.0)	0.309	0.174–0.549	
	Enough to keep (income > 20,001 baht)	55 (17.4)	18 (5.7)	37 (11.7)	0.418	0.204–0.859	
Stress/Anxiety	No	142 (44.8)	53 (16.7)	89 (28.1)			0.002 **
	Yes	175 (55.2)	38 (12.0)	137 (43.2)	2.147	1.309–3.521	

Chi squared test for nominal or ordinal variables and OR are reported. * Significant at *p* < 0.05, ** significant at *p* < 0.01. LBP: low back pain, OR = odds ratio, CI = confidence interval. ^a^ Based on household consumption expenditures of southern Thai people at an average of 17,095 baht.

**Table 2 ijerph-19-10492-t002:** Working profile of LBP predictors among rubber farmers (*n* = 317).

Characteristics	Category	Total *n* = 317 (%)	Without LBP *n* = 91 (%)	With LBP *n* = 226 (%)	OR	95% Cl	*p*-Value
Working days off per week	No	7 (2.2)	2 (0.6)	5 (1.6)			0.085
	1–2 Days/week	251 (79.2)	79 (24.9)	172 (54.3)	0.871	0.165–4.586	
	≥3 Days/week	59 (18.6)	10 (3.2)	49 (15.5)	1.960	0.332–11.568	
Working hours per day	1–4 h/day	107 (33.8)	26 (8.2)	81 (25.6)			0.236
	5–8 h/day	186 (58.7)	60 (18.9)	126 (39.7)	0.674	0.394–1.155	
	9–12 h/day	24 (7.6)	5 (1.6)	19 (6.0)	1.220	0.414–3.591	
Break time in a day	No	53 (16.7)	11 (3.5)	42 (13.2)			0.161
	Yes	264 (83.3)	80 (25.2)	184 (58.0)	0.602	0.295–1.230	
Inheritance career	No	82 (25.9)	31 (9.8)	51 (16.1)			
	Yes	235 (74.1)	60 (18.9)	175 (55.2)	1.773	1.039–3.024	0.034 *
Overtime	No	227 (71.6)	71 (22.4)	156 (49.2)			0.108
	Yes	90 (28.4)	20 (6.3)	70 (22.1)	1.593	0.900–2.819	
Agricultural registration ^b^	No	53 (16.7)	22 (6.9)	31 (9.8)			
	Yes	264 (83.3)	69 (21.8)	195 (61.5)	2.006	1.088–3.697	0.024 *
Work without training (habitual work)	No	52 (16.4)	21 (6.6)	31 (9.8)			
	Yes	265 (83.6)	70 (22.1)	195 (61.5)	1.887	1.018–3.500	0.042 *

Chi squared test for nominal or ordinal variables and OR are reported. * Significant at *p* < 0.05. LBP: low back pain, OR = odds ratio, CI = confidence interval. ^b^ Rubber farmers who are registered with the Ministry of Agriculture and Cooperatives.

**Table 3 ijerph-19-10492-t003:** Ergonomics risk factor of LBP predictors among rubber farmers (*n* = 317).

Variables	Category	Total *n* = 317 (%)	Without LBP *n* = 91 (%)	With LBP *n* = 226 (%)	OR	95% Cl	*p*-Value
Repeating movement	No	33 (10.4)	13 (4.1)	20 (6.3)			
	Yes	284 (89.6)	78 (24.6)	206 (65.0)	1.717	0.815–3.617	0.152
Squatting	No	74 (23.3)	32 (10.1)	42 (13.2)			
	Yes	243 (76.7)	59 (18.6)	184 (58.0)	2.376	1.377–4.100	0.002 **
Kneeling	No	107 (33.8)	43 (13.6)	64 (20.2)			
	Yes	210 (66.2)	48 (15.1)	162 (51.1)	2.268	1.371–3.750	0.001 **
Heavy workload (Continue harvesting for ≥7 h)	No	108 (34.1)	44 (13.9)	64 (20.2)			
	Yes	209 (65.9)	47 (14.8)	162 (51.1)	2.370	1.433–3.918	0.001 **
Getting muscle tension from the body directly	No	111 (35.0)	40 (12.6)	71 (22.4)			
	Yes	206 (65.0)	51 (16.1)	155 (48.9)	1.712	1.038–2.824	0.034 *
Heavy lifting ^c^	No	115 (36.3)	40 (12.6)	75 (23.7)			
	Yes	202 (63.7)	51 (16.1)	151 (47.6)	1.579	0.960–2.598	0.071
Prolonged standing (8 h consecutively)	No	167 (52.7)	64 (20.2)	103 (32.5)			
	Yes	150 (47.3)	27 (8.5)	123 (38.8)	2.831	1.682–4.763	0.001 **
Exertion more than usual	No	187 (59.0)	61 (19.2)	126 (39.7)			
	Yes	130 (41.0)	30 (9.5)	100 (31.5)	1.614	0.969–2.687	0.065
Awkward posture ^d^	No	203 (64.0)	58 (18.3)	145 (45.7)			
	Yes	114 (36.0)	33 (10.4)	81 (25.6)	0.982	0.592–1.629	0.943

Chi squared test for nominal or ordinal variables and OR are reported. * Significant at *p* < 0.05, ** significant at *p* < 0.01. LBP: low back pain, OR = odds ratio, CI = confidence interval. ^c^ Lifting and moving the latex buckets weighing ≥ 10–60 kg continuously for 2–3 h. ^d^ Positions of the body that deviate greatly from the neutral position during work activities.

**Table 4 ijerph-19-10492-t004:** Multivariate analyses of risk factors associated with LBP of rubber farmers.

	Multivariate Analysis				3-Marker Model								4-Marker Model			
	B	*p*-Value	AOR [95% CI]	R^2^	B	*p*-Value	AOR [95% CI]	R^2^	B	*p*-Value	AOR [95% CI]	R^2^	B	*p*-Value	AOR [95% CI]	R^2^
Sociodemographic/individual predictors																
Working experience (years)	0.418	0.144	1.519 [0.867–2.662]	0.216	0.555	0.037 *	1.743 [1.034–2.937]	0.100								
Sufficient income ^a^	0.870	0.008 **	0.419 [0.220–0.799]		0.935	0.002 **	0.393 [0.215–0.718]									
Stress/Anxiety	0.228	0.467	1.256 [0.680–2.319]		0.426	0.138	1.531 [0.872–2.690]									
Working profile predictors																
Inheritance career	0.477	0.119	1.612 [0.885–2.936]						0.505	0.070	1.657 [0.960–2.861]	0.057				
Agricultural registration ^b^	0.797	0.020 **	2.218 [1.132–4.346]						0.704	0.028 *	2.022 [1.078–3.792]					
Work without training (habitual work)	0.676	0.054	1.966 [0.990–3.904]						0.712	0.027 *	2.037 [1.083–3.832]					
Ergonomics risk predictors																
Squatting	0.285	0.413	1.329 [0.672–2.630]										0.558	0.082	1.748 [0.932–3.276]	0.125
Heavy workload(Continue harvesting for ≥7 h)	0.553	0.061	1.739 [0.976–3.099]										0.752	0.006 **	2.120 [1.242–3.621]	
Get muscle tension from the body directly	0.510	0.144	0.600 [0.303–1.190]										0.421	0.208	0.656 [0.341–1.264]	
Prolonged standing (8 h consecutively)	1.081	0.001 **	2.948 [1.546–5.618]										1.080	0.001 **	2.944 [1.586–5.465]	

* Significant at *p* < 0.05, ** significant at *p* < 0.01, LBP: low back pain, OR = odds ratio, AOR = adjusted odds ratio, CI = confidence interval, B = beta coefficient, and R^2^ = Nagelkerke determination coefficient. ^a^ Based on household consumption expenditures of southern Thai people at an average of 17,095 baht. ^b^ Rubber farmers who are registered with the Ministry of Agriculture and Cooperatives.

## Data Availability

Not applicable.

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
