# Peer review of "Predictors of Low Back Pain Risk among Rubber Harvesters"

_ijerph, 2022, doi:10.3390/ijerph191710492_

Round 1

Reviewer 1 Report

Main concerns:

1.     Lack of information about the selected factors for investigation.

2.     The definitions of the categories of many variables are not clear or too subjective. For example, how to determine the “Sufficient income”, what means for enough not left, enough keep…?   

3.     The factor of income, working time, workload and work stress are correlation to each other. It will be better to discuss the point due to workload is identified as a risk factor for LBP.

4.     Some results are not taken up in the discussion. In part, these would contradict the interpretation described, so that it should be checked again overall whether the results described are also discussed in their entirety. For example, the relationship between agricultural registration and LBP.

5.     The discussion section needs a heavy revision to make deep discussion. Most of the findings are consistent with previous studies. What is the new insight from the study? Moreover, the reference (Udom, C.; Janwantanakul, P.; Kanlayanaphotporn, R. The prevalence of low back pain and its associated factors in Thai 352 rubber farmers. J Occup Health. 2016, 58, 534–542. DOI:10.1539/joh.16-0044-OA.) seems very similar to the current study. However, none of paragraph to discuss the difference between the Udom et al’s work and current study). By the way, the predictors factors pointed by Udom et al. (2016) were totally differed to current study. Why?

6.     The analyses are not comprehensible without characteristic values. The AOR and the CI alone are not sufficient for understanding the analyses. For example, the value of B and R² Nagelkerke of the bivariate /multivariate logistic regression of each variable need to be presented. Please see the following references and consider to cite.

-Cabral et al., 2019. Is physical capacity associated with the occurrence of musculoskeletal symptoms among office workers? A cross-sectional study. Int Arch Occup Environ Health 92(8):1159-1172.

- Hong X et al., 2022. Risk factors associated with musculoskeletal disorders symptoms among manual porcelain workers at different workstations. Int Arch Occup Environ Health. https://doi.org/10.1007/s00420-022-01879-z.

Moreover, many latest reference need to be cited. In current form is a little old.

7.     Although it is stated what is already known about this topic and the knowledge gap, the study hypothesis and contribution were not clearly outlined. Please, include these.

8.     The main finding of the manuscript is obtained the 5 significant risk predictors for LBP among rubber harvesters including working experience, agricultural registration, habitual work, heavy workload and prolonged standing. However, most of the above predictors are subjective feeling. The quantitative measures are less, only can see in prolonged standing factor has definition for standing more than 8h and the working experience. Especially for habitual work and heavy workload factors, how to determine it? How to use the findings for the piratical application? 

Minor comments: 

1.     Ln 16-17 , “…with a 100% response rate”. This is very impressive. How can investigator achieve the high response rate? Dose it has any process?

2.     Ln 33-35 here needs several references to support.

3.     Ln 47-49. The reference 14 is focused on bankers not farmer. The authors can not quote it to support the opinion here. 

4.     Ln 55-56 “… risk factors…”It will better to describe what are the risk factors.

5.   Ln 89-90 It will be better to give the detail factors in every category (ciodemographic/individual factors, working environment, and ergonomic factors.)

6.     Ln 98-99 Why apply the inclusion criteria in the study? (e.g., 2 years or more and 28 days before investigation). The reasons for the criteria need to be addressed well.

7.     Ln 105-107. What is the “specifically modified questionnaire”? Moreover, any reference to support the rule for LBP determination?

8.     Ln 122 the statistical software needs to state.

9.     Ln 124-125. Why OR<1 was considered as protective factor? Sometimes the categories of each dependent variable are decided by investigator. 

10.   Ln 176, 187 & 193. Here should set as a subsection.  

11.   For Table 3, a definition of the awkward posture is needed. What is the difference between awkward posture, kneeing, Squatting,Repeating movement, Heavy lifting and Prolonged  Standing…) 

Author Response

We thank the Reviewer for the comments and suggestions.  We have revised our manuscript accordingly. We have provided below point-by-point responses to issues/concerns raised.

Major points

Point 1: Lack of information about the selected factors for investigation.

 Response: We have explained the factors that were selected for investigation in lines 45-46 and 50-60, as quoted below.

Ln 45-46: Long labor hours with twists, and awkward postures were the main cause of LBP in Swedish farmers.

Ln 50-60: LBP in farmers may be influenced by a variety of individual and ergonomic factors.  The incidence of work-related musculoskeletal disorders, including LBP, is reportedly influenced by personal characteristics such as age, sex, marital status, education, job experience, body mass index (BMI), smoking, income, expenditures, and non-work-related exercise. Psychosocial factors, such as anxiety and stress, as well as break time during the workday also play a role. Because of the demand from the market, farmers have a heavier workload during the harvest. Therefore, they were more likely to be exposed to ergonomic problems such as kneeling, squatting, heavy lifting, holding the same posture for longer than 8 hours, prolonged walking and standing, expending more effort than usual, heavier workload, and the stress of movement directly on the body.

 Point 2: The definitions of the categories of many variables are not clear or too subjective. For example, how to determine the “Sufficient income”, what means for enough not left, enough keep…?

Response: We have explained "Sufficient income" in the table legend below Table 1. Sufficient income means income that is sufficient for household consumption expenditures in the southern Thailand area. Household consumption expenditures in the southern Thailand area are equal to 17,095 baht.

We have identified the income range of "Not enough", "Enough but not enough", and "Enough to keep" in Table 1 are as follows: Not enough (income 5,000-15,000 baht); Enough but not left (income >15,001-20,000 baht); and Enough to keep (income >20,001 baht).

Point 3: The factor of income, working time, workload and work stress are correlation to each other. It will be better to discuss the point due to workload is identified as a risk factor for LBP.

Response: We have added to the discussion by highlighting the point of heavy workload in lines 282-284 and 308-316 as quoted below.

Ln 282-284: These findings highlighted the heavy workload during the harvesting season that may lead to chronic tension in the musculoskeletal system.

Ln 308-316: Especially concern about low-income which people want to work more to earn more, resulting in LBP symptoms. From this study, it was found that most of the rubber farmers were in marital status (82.6%), with family burdens to take care, and earn money to support their family. When income is insufficient, it cuts extra rubber (28.4%) that very workload over time. Along with emotional stress, social stress, particularly concern about low income, will further increase pain as it motivates people to work harder for additional income. As a result, this type of stress results in LBP symptoms. Stress, anxiety, and strain on muscles triggered the secretion of lactic acid, causing fatigue and more severe LBP.

Point 4: Some results are not taken up in the discussion. In part, these would contradict the interpretation described, so that it should be checked again overall whether the results described are also discussed in their entirety. For example, the relationship between agricultural registration and LBP.

Response: We have rechecked the factors that weren't mentioned and added them to the discussion in lines 252-262, as quoted below.

Furthermore, most of the farmers in this study were inheritance careers (74.1%), worked without training (83.6%), having more than 20 years of work experience (26.2%). These supported the notion that those who work constantly for a long time are more prone to experience greater musculoskeletal pain [32]. However, this study did not discover any relationship between LBP and individual characteristics such as sex, age, education, marital status, BMI, smoking, and physical activity. In contrast, the previous report has found the association of these factors with the occurrence of WMSD [19]. This result show that occupational and other environmental factor were also likely contributing to the prevalence of WMSD, including LBP. The influence of individual factors on the occurrence of LBP remains to be investigated. The percentage of registered volunteers was found to be 83.3%.

Point 5: The discussion section needs a heavy revision to make deep discussion. Most of the findings are consistent with previous studies. What is the new insight from the study? Moreover, the reference (Udom, C.; Janwantanakul, P.; Kanlayanaphotporn, R. The prevalence of low back pain and its associated factors in Thai 352 rubber farmers. J Occup Health. 2016, 58, 534–542. DOI:10.1539/joh.16-0044-OA.) seems very similar to the current study. However, none of paragraph to discuss the difference between the Udom et al’s work and current study). By the way, the predictors factors pointed by Udom et al. (2016) were totally differed to current study. Why?

Response: We thank the Reviewer for the suggestions. We have discussed this point further in lines 268-275, as quoted below.

In addition, our findings show the same trend as previous studies reported that 35% of rubber farmers who were working experience one year have LBP symptoms at least 24 hours of pain duration. However, the reported factors are different. We found that working experience, sufficient income, agricultural registration, work without training, heavy workload and prolonged standing presented as risk predictors for LBP, while they showed an active job, history of LBP, and current work position can affect LBP.  It is possible that our study was interested in predicting LBP incidence related to the harvesting season.  Moreover, our study focused on the work profile and ergonomics factors.

Point 6: The analyses are not comprehensible without characteristic values. The AOR and the CI alone are not sufficient for understanding the analyses. For example, the value of B and R² Nagelkerke of the bivariate /multivariate logistic regression of each variable need to be presented. Please see the following references and consider to cite.

-Cabral et al., 2019. Is physical capacity associated with the occurrence of musculoskeletal symptoms among office workers? A cross-sectional study. Int Arch Occup Environ Health 92(8):1159-1172.

- Hong X et al., 2022. Risk factors associated with musculoskeletal disorders symptoms among manual porcelain workers at different workstations. Int Arch Occup Environ Health. https://doi.org/10.1007/s00420-022-01879-z.

Moreover, many latest reference need to be cited. In current form is a little old.

Response: We have added the value of B and R² Nagelkerke to Table 4 as shown below.

Multivariate analysis

3-marker model

4-marker model

B

p-value

AOR [95% CI]

R2

B

p-value

AOR [95% CI]

R2

B

p-value

AOR [95% CI]

R2

B

p-value

AOR [95% CI]

R2

Sociodemographic/ individual predictors

Working experience (years)

0.418

0.144                    

1.519 [0.867-2.662]

0.216

0.555

0.037*

1.743

[1.034-2.937]

0.100

Sufficient incomea

0.870

0.008**

0.419

[0.220-0.799]

0.935

0.002**

0.393 [0.215-0.718]

Stress/Anxiety

0.228

0.467                                   

1.256 [0.680-2.319]

0.426

0.138

1.531 [0.872-2.690]

Working profile predictors

Inheritance career

0.477

0.119                    

1.612 [0.885-2.936]

0.505

0.070

1.657 [0.960-2.861]

0.057

Agricultural registrationb

0.797

0.020**

2.218 [1.132-4.346]

0.704

0.028*

2.022 [1.078-3.792]

Work without training (habitual work)

0.676

0.054

1.966 [0.990-3.904]

0.712

0.027*

2.037 [1.083-3.832]

Ergonomics risk predictors

Squatting

0.285

0.413                    

1.329 [0.672-2.630]

0.558

0.082                    

1.748 [0.932-3.276]

0.125

Heavy workload

(Continue harvesting for ≥7 hours)

0.553

0.061                    

1.739 [0.976-3.099]

0.752

0.006**

2.120 [1.242-3.621]

Get muscle tension from the body directly

0.510

0.144                    

0. .600 [0.303-1.190]

0.421

0.208                    

0.656 [0.341-1.264]

Prolonged standing (8 hours consecutively)

1.081

0.001**                

2.948 [1.546-5.618]

1.080

0.001**                

2.944 [1.586-5.465]

*Significant at p<0.05, **significant at p<0.01, LBP: low back pain, OR = odds ratio, AOR = adjusted odds ratio, CI = confidence interval, B =beta coefficient, and R2 =Nagelkerke determination coefficient. a Based on household consumption expenditures of southern Thai people at an average of 17,095 baht.  b Rubber farmers who are registered with the Ministry of Agriculture and Cooperatives.

We have added the suggested reference to the list.

Point 7: Although it is stated what is already known about this topic and the knowledge gap, the study hypothesis and contribution were not clearly outlined. Please, include these.

Response: We have added the hypothesis of this study in lines 95-97, as quoted below.

 We hypothesized that the subject characteristics, working profile, and ergonomic risk variables could predict the trends of symptom severity on LBP occurrence among rubber farmers.

Point 8: The main finding of the manuscript is obtained the 5 significant risk predictors for LBP among rubber harvesters including working experience, agricultural registration, habitual work, heavy workload and prolonged standing. However, most of the above predictors are subjective feeling. The quantitative measures are less, only can see in prolonged standing factor has definition for standing more than 8h and the working experience. Especially for habitual work and heavy workload factors, how to determine it?

How to use the findings for the piratical application?

Response: We have explained "Agricultural registration" in the table legend below Table 2 and Table 4. Agricultural registration means rubber farmers who are registered with the Ministry of Agriculture and Cooperatives.

We have identified the definition of habitual work in Table 2 and Table 4. Habitual work is work without training.

We have also inserted the definition of heavy workload in Table 3 and Table 4. Heavy workload means continuing to harvest for ≥7 hours.

Moreover, we have identified heavy lifting in the table legend below Table 3. Heavy lifting means lifting and moving the latex buckets weighing ≥10-60 kg continuously for 2-3 hours.

We have explained the application of this finding in lines 348-350, as quoted below.

The findings of this study should be used as information to train and educate about the appropriate work procedures for the rubber harvesting process.  Furthermore, such predictive factors should be further explored for disease prevention.

Minor points

Point 1: Ln 16-17 , “…with a 100% response rate”. This is very impressive. How can investigator achieve the high response rate? Dose it has any process?

Response: We have done the recruitment process through this process.

First, we promoted the project details through village monthly meetings.

Second, we asked the village health volunteers for help. 

Finally, we provided a comfortable place located close to the volunteer's home during the data collection. We found that they showed an unbiased interest in participating in this project.

Point 2: Ln 33-35 here needs several references to support.

Response: We have added references as shown.

[4] Kolstrup, C.L. Work-related musculoskeletal discomfort of dairy farmers and employed workers. Occupational Medicine and Toxicology. J Occupational Medicine and Toxicology. 2012, 7. DOI:10.1186/1745-6673-7-23.

[5] Sarker, A.H.; Islam, M.S.; Haque, M.M.; Parveen, T.N. Prevalence of Musculoskeletal Disorders among Farmers. Orthopedics & Rheumatology. 2016, 4. DOI: 10.15406/mojor.2016.04.00125.

[6] NG, Y.G.; Tamrin, S.B.M.; YIK, W.M.; Syah, I. Prevalence of Musculoskeletal Disorder and Association with Productivity Loss: A Preliminary Study among Labour Intensive Manual Harvesting Activities in Oil Palm Plantation. The Public Health J Burapha University: Industrial Health. 2017, 52, 78–85.

Point 3: Ln 47-49. The reference 14 is focused on bankers not farmer. The authors can not quote it to support the opinion here. 

Response: We thank the Reviewer for the suggestion.  We have changed the reference to

[18] Kongtawelert, A.; Buchholz, B.; Sujitrarath, D.; Laohaudomchok, W.; Kongtip, P.; Woskie, S. Prevalence and factors associated with musculoskeletal disorders among Thai Burley Tobacco farmers. Int J Environ Res Public Health. 2022, 19, 6776. DOI:10.3390/ijerph19116779.

Point 4: Ln 55-56 “… risk factors…”It will better to describe what are the risk factors.

Response: We thank the Reviewer for the suggestion. We have given an example of risk factors in line 67-69, as quoted below.

Furthermore, little is known regarding the job profile, ergonomics, and psychological risk factors for LBP among rubber farmers.

Point 5: Ln 89-90 It will be better to give the detail factors in every category (ciodemographic/individual factors, working environment, and ergonomic factors.)

Response: We have given the detail of every category in lines 103-112, as quoted below.

In this study, the risk factors for LBP were investigated, including sociodemographic/individual characteristics including those of sex, age, education, marital status, BMI, smoking, working experience (years), physical activity, income based on household consumption expenditures of southern Thai people, and stress/anxiety. Working profiles include working days off per week, working hours per day, break time in a day, inherited careers, overtime, agricultural registration with the Ministry of Agriculture and Cooperatives, work without training, and ergonomic factors like repeated movement, squatting, kneeling, heavy workload (continuous harvesting for over 7 hours), getting muscle strain from the body directly, heavy lifting (lifting and carrying latex buckets weighing approximately 10 and 60 kg continuously) were also investigated.

Point 6: Ln 98-99 Why apply the inclusion criteria in the study? (e.g., 2 years or more and 28 days before investigation). The reasons for the criteria need to be addressed well.

Response: We have added reference as below for 2 years or more.

[8] Intranuovo, G., Maria, L. D., Facchini, F., Giustiniano, A., Caputi, A., Birtolo, F., & Vimercati, L. (2019). Risk assessment of upper limbs repetitive movements in a fish industry. BMC Research Notes, 12, 354. doi:10.1186/s13104-019-4392-z.

 We found that occupations with two or more years of upper extremity work had a higher risk of pain than those who worked less than two years. We also added the detail of this factor in lines 35-37, as quoted below.

Compared to individuals who have worked for less than 2 years, employees who have worked for 2 years or more are more likely to develop LBP.

We have explained more about 28 days before investigation criteria in lines 120-122, as quoted below.

To obtain the most accurate information and minimize recall bias, these subjects carried on with their regular work activities for the 28 days prior to data collection.

Point 7: Ln 105-107. What is the “specifically modified questionnaire”? Moreover, any reference to support the rule for LBP determination?

Response: We have identified a questionnaire that was used in lines 128-129, as quoted below.

The Nordic Musculoskeletal Questionnaire (NMQ) was adopted to investigate low back pain.

We have added reference as below to the references list to support the use of the questionnaire.

[28] Cabral, A.M.; Moreira, R.F.C.; Barros, F.C.; Sato, T.O. Is physical capacity associated with the occurrence of musculo-skeletal symptoms among office workers? A cross-sectional study. Int Arch Occup Environ Health. 2019, 92, 1159-1172. DOI:10.1007/s00420-019-01455-y.

[29] Kuorinka, I.; Jonsson, B.; Kilbom, A.; Vinterberg, H.; Bierring-Sorensen, F.; Andersson, G. Standardized Nordic Ques-tionnaires for the Analysis of Musculoskeletal Symptoms. App Ergon. 1987, 18, 233-237. DOI:10.1016/0003-6870(87)90010-X.

Point 8: Ln 122 the statistical software needs to state.

Response: We thank the Reviewer for the suggestion. We have added the statistical program in lines 147-148, as quoted below.  

Statistical analysis was done using IBM SPSS Statistics28.0 (IBM Site Number:4225416).  

Point 9: Ln 124-125. Why OR<1 was considered as protective factor? Sometimes the categories of each dependent variable are decided by investigator.

Response: We thank the Reviewer for the suggestion. We have removed that sentence to avoid confusion.  We have added a reference list as below to support the interpretation of the odds ratio.

[31] McHugh ML. The odds ratio: calculation, usage, and interpretation. Biochem Med. 2009, 19, 120–126. DOI: 10.11613/BM.2009.011.

Point 10: Ln 176, 187 & 193. Here should set as a subsection. 

 Response: We have designated them as subsections as follows:

3.2.2 Three-marker model, line 201

3.2.3 Four-markers model, line 212

3.2.4 Potential Value of Risk predictors, 218

Point 11: For Table 3, a definition of the awkward posture is needed. What is the difference between awkward posture, kneeing, Squatting,Repeating movement, Heavy lifting and Prolonged  Standing…)

Response: We have added a definition of the awkward posture in the table legend below Table 3.  Awkward posture refers to positions of the body that deviate greatly from the neutral position during work activities.

Reviewer 2 Report

This is not a randomized cluster design since you recruited participants with the highest number of rubber farmers and allocated them to sub-groups with and without LBP

Any study does not support the validity and reliability of the questionnaire. Also, questionnaires for other outcome measures (inadequate income, stress, and anxiety) are not specified.

Confounders (height, body mass, body composition, work experience, physical activity...) are not specified in the Methods section, despite the potential confounding influence on study results. First, a preliminary analysis should be conducted to examine the confounding influence of all factors on LBP. If confounding influence exist (R2 >30%), multiple regression analyses should be conducted to assess the shared variance (R2) between LBP and specific factor while adjusting for confounders.

Working posture was analyzed in the Results section, although there is no information concerning measurement procedures in the Methods section.

Author Response

We thank the Reviewer for the comments and suggestions.  We have revised our manuscript accordingly. We have provided below point-by-point responses to issues/concerns raised.

Point 1: This is not a randomized cluster design since you recruited participants with the highest number of rubber farmers and allocated them to sub-groups with and without LBP

Response: We have corrected it in lines 116-118, as quoted below.

Using purposive random sampling, the three sub-districts with the highest number of rubber farmers were selected, then proportional sampling was based on the number of rubber farmers.

Point 2: Any study does not support the validity and reliability of the questionnaire.

Response: We have filled in the details of the questionnaire in lines 128-129, as quoted below.

The Nordic Musculoskeletal Questionnaire (NMQ) was adopted to investigate low back pain.

 In lines 132 and 133, we have described in detail the results of the quality check on the questionnaire, as quoted below.

The questionnaires were proved valid by 3 experts with an inspection content validity of 0.95 and were tested for reliability with Cronbach's alpha was 0.89.

Point 3: Also, questionnaires for other outcome measures (inadequate income, stress, and anxiety) are not specified.

Response: We have already detailed the factors in lines 103-112, as quoted below.

In this study, the risk factors for LBP were investigated, including sociodemographic/individual characteristics including those of sex, age, education, marital status, BMI, smoking, working experience (years), physical activity, income based on household consumption expenditures of southern Thai people, and stress/anxiety. Working pro-files include working days off per week, working hours per day, break time in a day, inherited careers, overtime, agricultural registration with the Ministry of Agriculture and Cooperatives, work without training, and ergonomic factors like repeated movement, squatting, kneeling, heavy workload (continuous harvesting for over 7 hours), getting muscle strain from the body directly, heavy lifting (lifting and carrying latex buckets weighing approximately 10 and 60 kg continuously) were also investigated.

Point 4: Confounders (height, body mass, body composition, work experience, physical activity...) are not specified in the Methods section, despite the potential confounding influence on study results.

Response: We have already given the details in lines 103-112 same as mentioned above. For weight and height, we present it in terms of BMI.

Point 5: First, a preliminary analysis should be conducted to examine the confounding influence of all factors on LBP. If confounding influence exist (R2 >30%), multiple regression analyses should be conducted to assess the shared variance (R2) between LBP and specific factor while adjusting for confounders.

Response: We effectively analyzed confounding influence and also tested the predictor model's credibility, as detailed below.

Lines 225-234; The explanatory power of the models for multivariate analysis was generally high, with pseudo R2 [Nagelkerke] values of 0.216. The explanatory power of the model for the three-marker model of sociodemographic/individual predictors and working profile predictors and the four-markers model was low with a pseudo R2 [Nagelkerke] value of 0.100, 0.057, and 0.125, respectively.

Besides, the results of Hosmer–Lemeshow test show that the p-values of the predicting models were 0.237 (Multivariate analysis), 0.609 (three-marker model of socio-demographic/individual predictors), 0.055 (three-marker model of working profile predictors), and 0.188 (four-marker model of ergonomics risk predictors) which indicates that the models are adequately calibrated.

Point 6: Working posture was analyzed in the Results section, although there is no information concerning measurement procedures in the Methods section.

Response: We have added more information on the ergonomics factors in lines 109-112, as quoted below.

……., and ergonomic factors like repeated movement, squatting, kneeling, heavy workload (continuous harvesting for over 7 hours), getting muscle strain from the body directly, heavy lifting (lifting and carrying latex buckets weighing approximately 10 and 60 kg continuously) were also investigated.

Reviewer 3 Report

This work is of interest in occupational health, but needs to be improved to give it scientific consistency.

There is a need for better coherence between the objective and the conclusions.

It is important to differentiate the results obtained between men and women. The gender variable is essential

It is necessary to provide or clearly specify the questionnaires used and whether they are validated questionnaires.

Please find the additional comments in the attachment.

Author Response

We thank the Reviewer for the comments and suggestions.  We have revised our manuscript accordingly. We have provided below point-by-point responses to issues/concerns raised.

Point 1: The objective should include prevalence of low back pain, differences by sex, causal risk factors and preventive attitudes in occupational hazards. It should be related to the final conclusions of the work.

Response: We have added the prevalence of LBP in the objectives, however, this study did not find a gender effect on LBP incidence in lines 94-95, as quoted below.

Therefore, this study aimed to investigate the prevalence of LBP and its risk factors among rubber farmers during the harvesting process.

Point 2: There is a need for better coherence between the objective and the conclusions.

 Response: We have also modified the conclusion of the study to make it more in accordance with the objectives in lines 342-350, as quoted below.

This study showed that LBP is a common work-related health problem among rubber farmers during harvesting.  The three-marker risk model shows that working experience, agricultural registration, and working without training factors are risk predictors of LBP.  The five-marker risk model showed the influence of ergonomics factors, especially heavy workload and prolonged standing on the severity of LBP.  Predicting risk factors for LBP may be a process in developing strategies for the prevention and control of WMSDs in this occupation.  The findings of this study should be used as information to train and educate about the appropriate work procedures for the rubber harvesting process.  Furthermore, such predictive factors should be further explored for disease prevention.

Point 3: Study Design and Setting It is important to incorporate a gender-independent view from the design at age, since the resulting damages can and should be different for men and woman. It is important to differentiate the results obtained between men and women. The gender variable is essential

Response: We thank the Reviewer for the suggestions. We have also been concerned about the effect of gender on LBP, but in this group, no such trend was observed.

Point 4: It is necessary to provide or clearly specify the questionnaires used and whether they are validated questionnaires.

Response: We have filled in the details of the questionnaire in lines 128-129, as quoted below.

The Nordic Musculoskeletal Questionnaire (NMQ) was adopted to investigate low back pain. 

 In lines 132 and 133, we have described in detail the results of the quality check on the questionnaire, as quoted below.

The questionnaires were proved valid by 3 experts with an inspection content validity of 0.95 and were tested for reliability with Cronbach's alpha was 0.89.

Point 5: The results obtained for men and women should be separated in all tables so that they can be compared at the end of the work and lead to more accurate conclusions.

Response: We thank the Reviewer for the suggestion to consider gender implications on LBP. The previous report may have identified a correlation. However, we did not observe any gender differences in the incidence of LBP in this group. We will take on the suggestion in our future works.

Point 6: work experience and all the factors considered in the study should already be included in the methodology as study variables and therefore be analyzed in the results. It should also be included whether or not the workers had received training on the occupational hazards of their work and especially on load handling and ergonomics.

Response: In lines 103-112, we have provided the specifics of the investigated factors, as quoted below.

In this study, the risk factors for LBP were investigated, including sociodemographic/individual characteristics including those of sex, age, education, marital status, BMI, smoking, working experience (years), physical activity, income based on household consumption expenditures of southern Thai people, and stress/anxiety. Working pro-files include working days off per week, working hours per day, break time in a day, inherited careers, overtime, agricultural registration with the Ministry of Agriculture and Cooperatives, work without training, and ergonomic factors like repeated movement squatting, kneeling, heavy workload (continuous harvesting for over 7 hours), getting muscle strain from the body directly, heavy lifting (lifting and carrying latex buckets weighing approximately 10 and 60 kg continuously) were also investigated.

We have listed the proportion of farmers working without training in Table 2.

Point 7: the conclusion should be modified according to the objectives of the study: prevalence, risk factors, gender differences in the results and impact of the variables analyzed on the presence of low back pain.

Response: We have revised the conclusions to align with the objective in lines 342-350, as quoted below.

This study showed that LBP is a common work-related health problem among rubber farmers during harvesting.  The three-marker risk model shows that working experience, agricultural registration, and working without training factors are risk predictors of LBP.  The five-marker risk model showed the influence of ergonomics factors, especially heavy workload and prolonged standing on the severity of LBP.  Predicting risk factors for LBP may be a process in developing strategies for the prevention and control of WMSDs in this occupation.  The findings of this study should be used as information to train and educate about the appropriate work procedures for the rubber harvesting process.  Furthermore, such predictive factors should be further explored for disease prevention.

Round 2

Reviewer 1 Report

All my comments have been carefully replied.  No more question is raised. 

Reviewer 3 Report

The authors have adequately revised the manuscript and corrected all the indications mentioned in the initial review. For my part, I consider that it can be published and that its content is of interest to occupational health professionals.

In future works it would be important to differentiate more clearly the results in men and women.